# Particulate Matter Exposure of Passengers at Bus Stations: A Review

**DOI:** 10.3390/ijerph15122886

**Published:** 2018-12-17

**Authors:** Le Thi Nhu Ngoc, Minjeong Kim, Vu Khac Hoang Bui, Duckshin Park, Young-Chul Lee

**Affiliations:** 1Department of BioNano Technology, Gachon University, 1342 Seongnam-Daero, Sujeong-Gu, Seongnam-Si, Gyeonggi-do 13120, Korea; nhungocle92@gmail.com (L.T.N.N.); hoangvu210190@gmail.com (V.K.H.B.); 2Korea Railroad Research Institute (KRRI), 176 Cheoldobakmulkwan-ro, Uiwang-si, Gyeonggi-do 16105, Korea; mjkim88@krri.re.kr

**Keywords:** particulate matter, bus station, personal exposure, ANN model, ANFIS model

## Abstract

This review clarifies particulate matter (PM) pollution, including its levels, the factors affecting its distribution, and its health effects on passengers waiting at bus stations. The usual factors affecting the characteristics and composition of PM include industrial emissions and meteorological factors (temperature, humidity, wind speed, rain volume) as well as bus-station-related factors such as fuel combustion in vehicles, wear of vehicle components, cigarette smoking, and vehicle flow. Several studies have proven that bus stops can accumulate high PM levels, thereby elevating passengers’ exposure to PM while waiting at bus stations, and leading to dire health outcomes such as cardiovascular disease (CVD), respiratory effects, and diabetes. In order to accurately predict PM pollution, an artificial neural network (ANN) and adaptive neuro-fuzzy inference systems (ANFIS) have been developed. ANN is a data modeling method of proven effectiveness in solving complex problems in the fields of alignment, prediction, and classification, while the ANFIS model has several advantages including non-requirement of a mathematical model, simulation of human thinking, and simple interpretation of results compared with other predictive methods.

## 1. Introduction

Recently, particulate matter (PM) pollution has become an important concern worldwide due to its negative health effects. PM is small enough to penetrate and deposit into many organisms of the body. When exposed to PM for a long time, people may acquire serious symptoms related to cardiovascular diseases (CVD) (e.g., heartbeat, arrhythmia, and vascular dysfunction), lung cancer, skin irritation, diabetes, and especially respiratory health effects [1,2,3,4,5]. In particular, with population growth and the rise of private vehicles leading to increased levels of air pollution in the city center, people have been encouraged to travel by public transport such as the bus and subway system. However, it has been demonstrated that both the inside and outside of the bus and subway system accumulate high PM levels, and so passengers might be exposed directly to PM during their trips by public transport [6,7].

Although the study of commuter exposure to air pollutants is not a new field of research, there has not been any focus on personal exposure assessments while passengers wait at bus stops. Several studies have identified the pollution at bus stations using various methods, and found that the PM_10_ and PM_2.5_ levels were too high compared with standard air quality guidelines in Europe, the Americas, and Asia [8,9,10]. For instance, Xu et al. (2015) determined that PM_10_ pollution is mainly attributed to diesel vehicle emissions (28%), crustal dust (26%), coal combustion (22%), and cement (4.9%) at bus stops in China. PM_10_ during rush hours (254 ± 128 μg/m^3^) was 2.5 times higher than that during the ambient 24 h (103 μg/m^3^) [6]. In a study by Moore et al. (2012), passengers waiting at bus stations in Portland, USA could be exposed to PM_10_ and PM_2.5_ at 25.00 and 21.97 μg/m^3^, respectively [7].

This present review provides an overview of PM pollution at bus station, focusing on the characteristics, health effects, and factors affecting PM as outlined in the following sections below: “Classification and sources of PM”, “Health problems caused by PM exposure”, and “Factors affecting PM pollution at bus stations”, respectively. In addition, the personal exposure levels to pollutants around the world (Europe, the Americas, and Asia) are explicated in the section “Personal exposure to PM at bus stations”. Next, in the section “Future directions for reduction in personal PM exposure”, some PM-pollution predictive methods are proposed. Finally, a limitation and conclusions are given in the “Limitation of study” and “Conclusions” sections, respectively.

## 2. Classification and Sources of PM

PM is known as particle pollution, which contains micrometer-sized particles, including both inorganic and organic particles such as dust, soot, dirt, smoke, and liquid droplets [11,12,13]. Generally, PM originates from volcanoes, forest fires, dust storms, grassland fires, sea spray, and living vegetation. In addition, human activities including the burning of fossil fuels in traffic, industrial processes, and power plant operation also generate significant amounts of PM [11]. PM is composed of sulfur dioxide, elemental carbon known as black carbon, organic matter, and soot, all of which induce visual effects such as smog [11].

PM is classified based on its aerodynamic diameter, which is the main criterion for determining its ability to transport in air and penetrate the human body [14]. PM includes PM_10_, with an aerodynamic diameter 10 μm or less; fine aerodynamic particulate is defined as PM_2.5_ of 2.5 μm or less in diameter, and ultrafine particles (PM_0.1_) are categorized as extremely small, less than 0.1 μm in diameter [11,12,15]. The sizes of PM_10_ and PM_2.5_ can be compared to the diameters of fine beach sand (~90 μm) and human hair (~70 μm), respectively (Figure 1). PM_10_ (coarse PM) is known as inhalable particles that can penetrate into the respiratory tract (e.g., trachea, deep lungs, and bronchi) [14,16,17]. The coarse fraction of PM_10_ generally originates from construction and demolition operations, paved and unpaved roads, industrial processes, and agriculture as well as biomass burning [14,18]. On the other hand, PM_2.5_ can be emitted directly from the combustion of vehicles, industry, smokestacks, fires, and via atmospheric reactions of gases (e.g., NO_x_ and SO_2_) [12,14,18], PM_2.5_ can easily enter the alveolar region of the lung [16,17,19,20]. At bus stations, PM_2.5_ is particularly problematic, because waiting passengers can be easily exposed to this particulate as generated by the fuel burning process in vehicles (e.g., motorcycles, trucks cars, buses, and heavy-duty vehicles). [16]. According to a report of the World Health Organization (WHO) (2003) on PM in Europe, the annual average mass concentrations of both PM_10_ and PM_2.5_ are mainly contributed by sulfates, organic matter, nitrate, and black carbon [13].

Recent studies indicate that traffic, specifically wear and tear of vehicle components (e.g., tires and brakes) and road dust suspension, is one of the major sources of PM [19,21]. Indeed, in the big cities around the world, increased demand for transportation in the forms of cars, buses, and subways, has led to increased air pollution from vehicle emissions [22]. The Department of Transport in the United Kingdom (UK) reported a 21% increase in vehicle traffic from 2000 to 2010 (Figure 2) [22]. The wide variety of pollutants and PM emitted from these sources is primarily composed of volatile organic compounds (VOCs), nitrogen oxides (NO_x_), sulfur dioxide (SO_2_), carbon monoxide (CO), carbon dioxide (CO_2_), and metal particles [21,22,23].

Particularly, buses have been considered as an environmentally friendly form of transport relative to cars and other types. Buses utilize less fuel per person carried, thus producing less pollution than the number of either cars or motorbikes replaced [22,24]. Nevertheless, diesel engines used in buses emit large amounts of NO_x_, leading to larger emissions of black smoke and PM. Black smoke consists of numerous PM responsible for the soiling of buildings, and PM_2.5_ is correlated with a variety of adverse health effects [25].

## 3. Health Problems Caused by PM Exposure

The exposure effectiveness of PM depends not only on its chemical compositions and physical properties that can be influenced by local conditions including weather, seasons, sources of particles, and concentrations emitted [26], but also on human physical characteristics (e.g., breathing mode and volume of a typical person). Particle size is primarily responsible for the association between PM and human health problems: smaller particles can more easily penetrate into the human body and deposit deep into the respiratory tract [19,27]. Indeed, Atkinson et al. indicated that the cilia and mucus in nasal-breathing effectively filter only most particles greater than 10 μm in diameter; thus, PM_10_ and PM_2.5_ can easily infiltrate the body, settle down rapidly, and lodge in the bronchi or trachea (upper throat). However, the body may react to eliminate these intrusive PM via processes such as sneezing and coughing [28]. Londahl et al. acknowledged that particles less than 10 μm in diameter can enter the respiratory tract, starting from the nasal passages and proceeding into the alveoli and deep within the lungs, due to their excessive penetrability [29]. In addition, whereas particles in the range of 5–10 μm are most likely to be deposited in the tracheobronchial tree, those from 1 to 5 μm tend to be deposited in the alveoli and the respiratory bronchioles where gas exchange occurs [29,30] (Figure 3). During deposition in the lung, these particles may interfere with gas exchange and then escape into the bloodstream, resulting in significant health problems [19,30]. On the other hand, particles smaller than 1 μm behave like gas molecules, thus entering into the alveoli (deposited by diffusion forces), and move further into tissue and the circulatory system [31]. Generally, as the human body cannot prevent exposure or adversely effects by PM, people may experience several health problems including CVD, respiratory health effects, diabetes, and premature death [1,23]. The WHO (2016) estimates that PM pollution contributes to about 4.2 million premature deaths each year (16% of lung cancer deaths, 26% of respiratory infection deaths, 17% of ischemic heart disease and stroke deaths, and 25% of chronic obstructive pulmonary disease deaths), ranking it as the 14th leading cause of death worldwide [32,33,34].

### 3.1. Cardiovascular Diseases (CVD)

Cardiovascular diseases (CVD) are recognized as some of the leading causes of mortality and morbidity in the world [35,36]. Recently, many traditional CVD risk factors have been identified, such as high blood pressure, diabetes, physical inactivity, smoking, and especially air pollution [1,19,37]. Particularly, PM is an important CVD risk factor in cases where people are exposed over long durations, because it can easily penetrate and deposit deep into the organism through several pathways both direct and indirect (Figure 4) [1,2,38]. Via the direct pathway, these particles translocate into the bloodstream and remote specific target organs [38,39]. In this systemic circulation, reactive oxygen species and ion channel interference play an important role in affecting the heart and vasculature [40]. In contrast, indirectly, these particles induce pulmonary-mediated oxidative stress and inflammatory responses, resulting in a less acute and adverse effects after several hours and days of inhalation [38,39,40,41,42].

#### 3.1.1. Effects of Short-Term Exposure

Several studies have estimated that short-term PM exposure (~a few days) and PM increase of 10 μg/m^3^ are associated with increased relative risk (RR) of daily CVD mortality in both PM_10_ and PM_2.5_ of 0.6–1.8% and 0.6–1.3%, respectively [32,39]. In addition, these PM concentrations may present a major acute risk for elderly people and those with heart diseases [44]. Although the risk to an individual at any one time point may be small, the burden on public health is enormous. An increase in short-term PM exposure leads to tens of thousands of mortalities per year in the United States. According to the Nation Morbidity Mortality Air Pollution Study (NMMAPS) data, an increase of 10 μg/m^3^ in PM_10_ caused a 0.7% increase in cardiopulmonary mortality (95% confidence interval (CI), 0.2–1.2%) [45]. In another study on the 38 biggest of China’s cities by Yin et al., the effect of increase per 10 µg/m^3^ PM_10_ on deaths from cardiorespiratory diseases was 0.62% (95% CI 0.43–0.81%), compared with 0.26% (95% CI 0.09–0.42%) for other causes of mortality [46]. In another, smaller trial on 12,000 patients in Utah, Pope et al. determined that a 10 μg/m^3^ increase in PM_2.5_ resulted in a 4.5% (95% CI, 1.1–8.0%) increase in acute ischemic coronary events [44].

#### 3.1.2. Effects of Long-Term Exposure

The adverse effect of long-term exposure to PM has been identified as even more harmful than short-term exposure in the “Harvard Six Cities” study, which showed that living in the heaviest polluted cities increased the risk of CVD mortality by 30% [39]. In 2007, a women’s health initiative study estimated that long-term PM exposure leads to a 24% (95% CI, 9–41%) increase in CVD per PM_2.5_ increase of 10 μg/m^3^ [47]. In addition, PM pollution is associated with CVD morbidity, as observed by some studies focused on hospital admissions for CVD. According to the Medicare data for 204 U.S. cities, a rise of 10 μg/m^3^ in PM_2.5_ concentration results in increased hospitalization for cerebrovascular disease (+0.81%), ischemic heart disease (+0.44%), arrhythmias (+0.57%), peripheral arterial disease (+0.86%), and heart failure (+1.28%) [39,48,49].

### 3.2. Respiratory Effects

Exposure to PM has been proven to be associated with a variety of respiratory health effects including respiratory symptoms (cough, phlegm, and wheeze), bronchial hyper-reactivity, acute-phase reaction, respiratory infections, decreased lung growth in children, chronic loss of pulmonary function in adults, and premature mortality in patients with chronic lung disease [1,50]. PM’s respiratory mechanisms consist of pulmonary injury from free radical peroxidation, imbalanced intracellular calcium homeostasis, and inflammation injury (Figure 5) [1,4,5,32]. In fact, when it enters the body, PM can directly affect macrophages, in which the alveolus responds rapidly to inhaled PM as an initial innate immune response, and then produces nitrogen species, reactive oxygen species, and releases TNF-α and IL-1, thus resulting in epithelial cell apoptosis and inflammation [32,51,52]. Besides, induced mitochondrial fusion and mitochondrial lipid peroxidation in lung macrophages might be an important mechanism contributing to respiratory diseases caused by PM [52].

According to the report of the WHO, approximately 16% of lung cancer deaths, 11% of chronic obstructive pulmonary disease (COPD) deaths, and 13% of respiratory infection deaths are caused by exposure to air pollution [52]. In addition, an American Cancer Society cohort study of 1.2 million American adults for 26 years (from 1982–2008) showed that with a PM_2.5_ increase of 10 μg/m^3^ per day, lung cancer mortality increased by 15–27% [4]. In another study, Karakatsani et al. reported that a 10 μg/m^3^ increase the in previous-day PM_10_ concentration was positively correlated with a 1.06% increase in cough symptoms (95% CI: 1.01–1.11) [53]. Guo et al. (2018) determined that long-term exposure to PM_2.5_ was linked to reduction and faster decline of lung function and that it was also associated with a significant increase in COPD risk (1.39%; 95% CI, 1.24–1.56) in Taiwan [54]. Jo et al. evaluated the effect of PM on patients at a hospital in South Korea (between 2007 and 2010) in terms of respiratory diseases. At that time, for the mean daily PM_10_ and PM_2.5_ concentration of 49.6 ± 20.5 and 24.2 ± 10.9 mg/m^3^, the mean numbers of acute bronchitis, allergic rhinitis, and asthma cases were 5.8 ± 11.9, 4.4 ± 6.1, and 3.3 ± 3.3, respectively [55].

### 3.3. Diabetes

Recently, at least five publications have identified the relationship between air pollution, especially PM, and diabetes [56,57,58,59,60], and showed that the main mechanism of PM’s increased incidence of diabetes is significantly involved in endothelial and mitochondrial dysfunction and inflammation of visceral adipose tissues [3]. In particular, the long-term exposure of adipose tissue macrophages to PM was characterized by increased IL-6 and TNF-α as well as reduced expression of IL-10, thereby promoting the production of innate immune cells in adipose tissue, which is pathophysiologic of type 2 diabetes [3,61]. In addition, this prolonged exposure also reduces inter-scapular brown adipose tissue (BAT) and mitochondrial size, which effects are accompanied by increases in nitrosative and oxidative stress in BAT, in combination with antioxidant gene induction including nicotinamide adenine dinucleotide phosphate (NADPH) quinone oxidoreductase 1, NF-E2-related factor 2, and glutamate-cysteine ligase modifier subunit, resulting in the downregulation of insulin in adipose gene profiles and reduction of uncoupling protein expression, all of which are risk factors for development of type 2 diabetes [62].

In fact, He et al. reported that PM_2.5_ was 1.25 % positively correlated with incidence of type 2 diabetes mellitus over a long-term exposure period (95% CI, 1.10–1.43). Therefore, with a 10 μg/m^3^ increase in PM_2.5_ per day, the incidence of type 2 diabetes would increase by 25% [63]. In a systematic review by Liang et al. (2014), PM_2.5_ was closely related to blood pressure variation of 1.39 mmHg (95% CI, 0.87–1.91) per 10 μg/m^3^ increase in PM_2.5_ [57]. In addition, in another study, Hansen et al. (2016) found that PM_10_ and PM_2.5_ were potential risk factors for diabetes development. The incidence of diabetes increased by 1.06% (95% CI, 0.98–1.14) and 1.11% (95% CI, 1.02–1.22) for each increase of 2.8 μg/m^3^ in PM_10_ and 3.1 μg/m^3^ in PM_2.5_, respectively [56].

## 4. Factors Affecting PM Pollution at Bus Stations

### 4.1. Meteorological Factors

Several environmental factors have been identified during PM_10_ and PM_2.5_ monitoring at bus stations. The association between PM, airborne particles, and meteorological parameters (e.g., wind speed, temperature, relative humidity, pressure, rain volume, and cloudiness) have been investigated in several recent studies [64,65,66,67,68,69].

Akyuz et al. studied the meteorological dependence of PM concentrations in winter and summer periods according to Pearson’s correlation analysis in Zonguldak, Turkey [65]. There were significant differences in the seasonal variations of PM_10_ and PM_2.5_ concentrations. In particular, the maximum daily PM_10_ and PM_2.5_ concentrations reached 66.7 μg/m^3^ and 32.4 μg/m^3^ in summer, and 116.7 μg/m^3^ and 83.3 μg/m^3^ in winter, respectively. The authors pointed out that atmospheric pressure indirectly influences pollutant concentrations by affecting atmospheric stability conditions. Indeed, high atmospheric pressure leads to low wind speed and stable stratification, which limit the spread of pollutants within the atmosphere [65].

Tecer et al. indicated that temperature has a significant negative effect on PM concentration with respect to the occurrence of episodic events in Turkey. The concentrations of PM_10_ and PM_2.5_ increased 7 and 6 times at the lowest temperature (10 °C), respectively [64]. They also reported that the wind speed increased from 1.39 to 2.80 m/s, resulting in an increase in pollution levels per day. Besides, when wind speed is too high, PM can be transported from nearby sources by the dilution effects of the wind, which indeed plays an important role in PM movements [64].

In another study, this one by Unal et al., variations of PM_10_ concentrations as influenced by meteorological factors including wind direction, wind speed, and high pressure in Istanbul were analyzed [68]. Figure 6 showed that with increases of wind speed, the average concentration of PM_10_ was higher than 50 μg/m^3^ and typically highest with winds blowing in the south-west (SW) and east-north-east (ENE) directions. They also demonstrated that a high pressure system can induce light wind and stable atmospheric conditions, in consequence of which, the highest PM_10_ level was found, and vice versa for strong wind and unstable atmospheric conditions [68].

Fondelli et al. studied PM_2.5_ concentrations during working days and in heavy traffic to predict ambient pollution levels in Florence, Italy [67]. According to data from the Tuscan Environmental Protection Agency (ARPAT) for 24-h PM_2.5_ measurements, they analyzed and estimated that PM_2.5_ hourly averages were closely correlated with pressure, precipitation, wind speed, and wind direction. In this study, low wind speeds (average below 2 m·s^−1^) effected increases in PM_2.5_ mass and composition [67].

### 4.2. Traffic Factors

Among the environmental factors influencing PM concentrations, traffic is the major one at bus stations [10,66,70]. According to the report of the Air Quality Expert Group (AQEG) (2012) in the UK, road traffic contributes significantly to the existence of PM_2.5_ in ambient air pollution [71]. Indeed, traffic is closely associated with PM_2.5_ pollution via two pathways including exhaust particles produced from gasoline- and diesel-engine vehicles (e.g., buses, private vehicles, and motorcycles) and non-exhaust particles originating from various physical processes (e.g., tire and brake wear, tire abrasion of road surface, and blowing off of dust particles caused by vehicle motion) [10].

Moreover, the internal fossil fuel combustion of diesel buses is known to be the main mobile source of PM emissions at bus stations. Bus types, traffic volume, and the presence of cigarette smoking at bus shelters also are factors [66]. In addition, the location of bus shelters is one of the contributors to personal exposure to PM. Most bus shelters are usually located on main roads or near intersections; thus, particles originally exhausted from other private vehicles will easily accumulate, especially during rush hours or green lights near bus stops, when vehicle flows are high [10,66]. Consequently, passengers waiting at bus shelters can be exposed directly to large amounts of PM. Moreover, Hess et al. (2010) evaluated the role of individual cigarette smoking on PM pollution at bus stations, finding that the presence of cigarette smoking at the level of 0.01 increased the PM_2.5_ exposure of waiting passengers by 22.74 μg/m^3^ [66].

The effects of traffic factors on PM concentration have been determined by various methods. Zhang and Batterman (2010) used generalized additive models to estimate vehicle contributions to PM pollution near roadways [72]. They indicated that traffic count was in a positive relationship with PM_2.5_ concentration rise at points near the road (Figure 7).

In addition, PM pollution is also correlated with vehicle flow [21,73]. In an investigation by Moore et al. (2012), averages of 1267 and 1415 vehicles per hour at bus shelters in the morning and evening, respectively, were positively associated with PM_10_ and PM_2.5_ pollution in Powell Boulevard, UK. Besides, the authors pointed out that the orientation of bus shelters also influenced the distribution of PM inside and outside. They observed that there were higher PM concentrations inside than outside of shelters oriented toward the roadway. Contrastingly, the concentrations were higher outside than inside of shelters oriented away from the roadway [73].

## 5. Personal Exposure to PM at Bus Stations

### 5.1. PM Exposure Levels in Europe

In the 1999–2000 period, due to excessive levels of PM in London, UK, Adams et al. conducted a comprehensive PM_2.5_ personal exposure study of users of bus, car, and bicycle transport [74]. It was estimated that passengers during bus transit and while waiting at bus stops were exposed to 34.0 ± 1.8 μg/m^3^ and 30.9 ± 2.1 μg/m^3^ in the summer and winter, respectively. They also indicated that meteorological variables, traffic density, and route are all closely linked to individual exposure. Wind speed was found to be the most important factor influencing personal exposure, higher wind speeds leading to lower personal exposure levels. In particular, the difference in personal exposure was 1.5–2.0 times when comparing the 10th and 90th percentiles of wind speeds. In 2009, Kaur et al. calculated personal exposures to PM_2.5_ in Central London, UK by five transport models (walking, car, taxi, cycling, and bus) [75]. The analyses estimated that for an average temperature of 14 °C, 70% humidity, and a wind speed of 1 to 5.8 m/second, PM_2.5_ concentrations were obtained at an average of 34.1 ± 11.3 μg/m^3^ during waiting and transit by bus. In another study, Cevallos monitored PM_2.5_ pollution at seven bus stops on the campus of the University of Manchester, UK, and found that the concentration of pollutant was obtained in the range of 13.66–25.72 μg/m^3^ [10]. In addition, they reported that bus stop direction and design were correlated with local exhaust emissions and meteorological factors contributing to differences in PM levels inside and outside of bus stations [10].

Onat and Stakeeva (2013) assessed the personal exposure of commuters in public transport to PM_2.5_ in central Istanbul, Turkey. The highest average PM_2.5_ exposure of bus passengers was 120.4 ± 73.5 μg/m^3^ and 84.5 ± 42.8 μg/m^3^ during rush hours and non-rush hours, respectively [20]. Further, they evaluated wind speed, temperature, and relative humidity affecting PM_2.5_ concentration distribution. The PM_2.5_ concentration was positively associated with humidity and wind speed (0.70 for inside bus and bus shelter), and negatively correlated with temperature (temperature increased as particle concentration decreased).

Fondelli et al. (2008) evaluated urban fine particle exposure concentrations inside and outside of bus stations in Florence, Italy [67]. At fixed-site monitoring stations, the average PM_2.5_ concentration was 32 μg/m^3^ (in the range of 22–52 μg/m^3^). According to the report of time-microenvironment-activity-diary data, the average exposure of Florentines was about 12% of personal PM_2.5_ exposure.

### 5.2. PM Exposure Levels in Americas

Hess et al. (2010) evaluated waiting passengers’ exposure to PM_2.5_ inside and outside bus stations in Buffalo, New York by investigating 840 min of concurrent exposure [66]. They used a multivariate regression model to assess the relationship between PM_2.5_ exposure and three vectors of determinants including time and location, environmental factors, and physical setting and location. This model suggested that personal exposure to PM_2.5_ inside the bus shelters was 17.24 ± 16.60 μg/m^3^, higher than that outside by 14.72 ± 8.19 μg/m^3^, due to cigarette smoking. The cigarette smoking, which was found to increase PM_2.5_ exposure to 22.74 μg/m^3^, was the largest factor enhancing personal exposure while passengers wait at bus stations.

In addition, Moore et al. (2012) also conducted an empirical study of PM exposure for passengers waiting at bus stations in Portland, USA [7]. This study compared the personal exposure at two-sided bus shelters which either face roadway traffic or are oriented away from it. The mean values of PM_10_ and PM_2.5_ were 25.00 and 21.97 μg/m^3^ relative to independent variables (e.g., shelter orientation, vehicle flow, wind speed, wind direction, temperature, and humidity). Particularly, with an opening oriented towards the roadway, the PM concentration inside the bus shelter was higher than that outside. By contrast, the inside of shelters oriented away from roadway traffic had a lower PM concentration than did the outside. Besides, they reported that vehicle flow showed a significant correlation with PM concentration at bus stations.

The correlation of particulate air pollution at bus stops and vascular reactivity in Ottawa, Canada was demonstrated by Dales et al. [8]. In this study, which included 39 healthy volunteers waiting outside bus shelters for 2 h, the flow-mediated vasodilation (FMD) of the brachial artery increased with in PM_2.5_ exposure. The authors indicated that a 30 μg/m^3^ increase in PM_2.5_ led to a 0.48% reduction in FMD.

### 5.3. PM Exposure Levels in Asia

Velasco and Tan (2016) measured the exposure of passengers to particles while waiting at bus stops in the humid and hot weather of Singapore [9]. In this study, they used a set of portable battery-operated sensors to evaluate traffic particle concentrations at the bus stations. It was estimated that traffic exhaust particles at these bus stops contained mostly PM_1_, PM_2.5_, and PM_10_ ranging from 89% to 94% of total exhaust particles. Although waiting times at bus stations in Singaporean are generally short (average 20 min a roundtrip per day), commuters might be exposed to concentrations 1.5–3 times greater than reported by local authorities with an average PM_2.5_ exposure of 23–57 μg/m^3^.

In China, Xu et al. conducted an empirical study on individual and population intake fractions of PM at bus stops to estimate personal exposure as well as the correlation between diesel PM and emissions [6]. The PM pollution was mainly caused by diesel vehicle emission (28%), crustal dust (26%), coal combustion (22%), cement (4.9%), and other sources. In addition, the PM_10_ concentration at bus stations during rush hours, at 254 ± 128 μg/m^3^, was much too high, compared with ambient 24 h and intercity bus terminal PM concentrations of 103 and 80 μg/m^3^, respectively. They also pointed out that bus stations near roadsides could accumulate high particles, thus causing high level of exposure to direct vehicle emissions.

In addition to city center bus stops, intercity bus terminals can also accumulate high particle levels, especially in waiting areas. Cheng et al. determined short-term exposures to PM_10_ and PM_2.5_ for passengers at a Taipei bus terminal station [76]. This study estimated PM_10_ and PM_2.5_ levels of 74 ± 30.5 and 50.75 ± 26.25 μg/m^3^, respectively, which had been exhausted directly from cruising and idling buses and outside traffic surrounding waiting areas. In another study, Salama et al. (2017) assessed air quality in bus terminal stations in the Kingdom of Saudi Arabia by applying dust collection calibrated devices [77]. By this method, PM_10_, PM_4_, PM_2.5_, and PM_1_ levels were measured as 185.1 ± 21.5, 118.8 ± 18.4, 112.8 ± 13.3, and 85.5 ± 9.9 μg/m^3^ in the morning and 172.3 ± 4.8, 137.8 ± 18.2, 93.8 ± 10.23, and 99.2 ± 4.3 μg/m^3^ in the evening, respectively, as compared with Saudi Arabia’s standard air quality guidelines.

## 6. Future Directions for Reduction in Personal PM Exposure

### 6.1. Pollution Prevention and Control

PM emissions can be minimized by pollution prevention and control technologies. Some typical regulations and air quality management actions have been applied to reduce pollution from concerning sources (traffic, industry, and human activities). Pollution prevention includes choosing clean fuels (natural gas) instead of diesel fuel, whose substitution can reduce the formation of secondary particles (NO_2_ and NH_3_) in traffic emission; utilizing cleaner processes such as advanced coal combustion technologies (coal gasification and fluidized-bed combustion) that may lower concentrations of products by incomplete combustion, and replacing older devices with cleaner ones enables better burning and fewer PM emissions [78,79]. Particularly, in order to reduce personal exposure indoors and outdoors, especially at bus stations, passengers should avoid smoking in bus shelters, reduce traveling during rush hour, and avoid outdoor activities when pollution levels are high [80].

In addition, a variety of PM-removal technologies with different physical and economic characteristics have been applied, such as impingement separators and electrostatic precipitators. Impingement separators rely on the inertial properties of the particles to separate them from the carrier gas stream, thereby collecting medium-size and coarse particles, while electrostatic precipitators show a high efficiency in collecting PM_2.5_ with well-designed, well-operated, and well-maintained systems, that can remove PM by using an electrostatic field to attract particles onto their electrodes [79].

### 6.2. Forecasting of PM Pollution

In order to avoid high pollution exposure, commuters should check current and forecasted air quality levels before undertaking outdoor and public transport. Forecasting of pollution is an important tool for taking effective pollution control measures that can provide an early warning against harmful air pollutants [81]. Among these various predictive models, modes of artificial intelligence such as artificial neural network (ANN) and adaptive neuro-fuzzy inference systems (ANFIS) models are classic methods used to forecast air pollution quantitatively [81,82].

#### 6.2.1. Artificial Neural Network (ANN) Models

ANN is a powerful data modeling method with proven efficacy in solving complex problems in the fields of alignment, prediction, and classification [82]. A number of recent studies have used ANN models to predict hourly or daily PM concentrations with low error values [82,83,84,85]. It is a mathematical model based on a collection of artificial neurons connected or functionally-related to each other, which behave like neurons in the biological brain [83,85]. Generally, its architecture consists of arrangement of neurons within several layers including an input layer, hidden layers, and an output layer (Figure 8); all neurons in a layer are connected to all neurons in adjacent layers by synaptic weights acting as signaling coefficients on the corresponding connections [83,85,86]. In ANN models, signals are transmitted from the first layer (the input layer) to the final layer (the output layer); the connecting signals between artificial neurons are real numbers, and the output of each of the artificial neurons is analyzed by a number of linear or non-linear statistical techniques [83,87]. In addition, ANN models have been developed on two popular neural network architectures including the Multilayer Perceptron (MLP) neural network and the Radial Basis Function (RBF) network.

##### Multilayer Perceptron Neural Network (MLP)

MLP, along with feed-forward network (FFN), is the most popular neural network architecture, and is used to analyze the relationships between different variables and to predict the outcome of response variables. Used when the number of independent variables is greater than one, the MLP model with the given i observation is shown in Equation (1),
(1)$$Y\;=\;\beta_{0}\;+\;\sum_{i=1}^{n}\beta_{i}X_{i}\;+\varepsilon_{i}$$
where X_*i*_ is the value of the input variable I, Y is the prediction calculated by a linear combination, the constant β_0_ and the regression coefficients β_*i*_ being computed by ordinary least-squares equation, and ε is a residual error [83].

The MLP model consists of three layers of neurons, each of which uses a linear combination function, and the input signal is processed in several steps in one direction: the input variables activate the first layer (input layer), the signal is then transmitted to the second layer (hidden layer), where this signal is elaborated, and finally, the activation states of the second layer are then passed to the last layer [82,83]. In detail, the number of neurons in the first layer corresponds to the number of input variables, the number of nodes in the second layer is chosen to limit the total number of weights that can avoid an important problem known as overtraining of the network, and the last layer includes a single neuron representing the output of the network [83].

Supervised training algorithms such as backpropagation (BP) play an important role in the ANN network. With BP, also known as “training”, repetitive inputs are presented to the neural network, the output is compared to the desired output, and an error is evaluated. After BP, the neural network adjusts the weights to reduce the error with each iteration and to improve outcomes closer and closer to the desired output. Indeed, this training is relatively easy and provides good support for predictive applications [82].

In fact, in a study of Feng et al. (2015), the MLP type of back-propagation neural network was applied to forecast PM_2.5_ pollution based on respective pollutant predictors as well as meteorological forecast variables as input data [88]. In this study, 85% of data was set for training, and the remaining data was used for testing. There were 10 variables in the first layer, as shown in Figure 9 (PM concentration, temperature max and min, wind, humidity, neighbor weighted, general condition, month of year, and day of week), and the prognostic predictors (wind, temperature, and humidity) were extracted from the predicted values published by meteorological authorities instead of real-time values. The hidden layer of this neural network contained 8 neurons that can obtain the best validation data for an optimal MLP structure. It has been reported that with the high daily level of PM_2.5_ in China, applications of this neural network can achieve an average accurate prediction of 90% compared to real-time values of PM_2.5_ concentrations prevalent at the same time.

Furthermore, Bianofiore et al. developed a recurrent architecture based on the MLP model for analysis and forecast of PM_10_ and PM_2.5_ [83]. In this model, the input layer was set with the activation state of the nodes in the intermediate layers, the processed signal was then transmitted back to the input level, and the neurons in the second layer contained the compressed information on the meteorological and chemical parameters of the previous time step, and so this architecture displayed a dynamic memory of the types of events provided as the inputs of the network. Meteorological conditions, real-time PM_10_ and PM_2.5_ concentrations from training data, and CO concentrations were used as the inputs to simulate PM_10_ and PM_2.5_ concentrations via the recurrent neural network. It was determined that this model is a potential tool for obtainment of the real-time PM_10_ and PM_2.5_ concentration information.

##### Radial Basis Function (RBF) Neural Network

Radial basis function (RBF) also is an effective feed-forward neural network. Although less well-known than the MLP model, the main advantages of the RBF network are its affording of the minimum approximating error of any function as well as the global optimum, and its generally much faster training [83,89,90,91]. Therefore, it has been widely used in a considerable number of applications (e.g., classification, regression problems, function approximation, prediction, and signal processing) with good results.

The RBF network consists of three different layers (Figure 10). The first layer has the same structure and function as the input layer of the MLP network, including various input neurons connected to each input, and the input neurons then feed the values to each of the neurons in the second layer (hidden layer). The main difference between RBF and other types of neural network is the hidden layer [83,91]. The hidden layer is a center nonlinear function that is symmetric to the local distribution, and includes a width parameter and a center position with the RBF based on the Gaussian distribution function (Equation (2)) [90,91]:(2)$$\emptyset_{j}(\chi )\;=\;\exp\;(-\frac{||\chi -{µ}_{j}||{}^{2}}{2\sigma_{j}^{2}})$$
where ∅_*j*_ is the nonlinear function of unit *j* in the RBF, χ is the input data vector, µ_*j*_ is the center of unit *j* in the RBF unit, and σ_*j*_ is the spread of the Gaussian basis function with respect to unit *j*.

The third layer of the RBF network is connected by linear units and creates output [83,89]. The extracted signal from the hidden layer is processed and the problem is solved by learning processes in the output layer. There are two sets of weights measuring the distance concerning input data and the output layer. In the first set of weights, the radial distance is computed for every unit between the non-linear inputs and the center of the basis function using the Euclidean distance algorithm, while the second weight is combined with the activation function known as the RBF, which generates outputs in the linear form [91]. The output can forecast pollutant concentrations following Equation (3):(3)$$Y_{k(x)}\;=\;\sum_{j=1}^{M}W_{kj}\emptyset_{j}\left(X\right)\;+\;W_{k0}$$
where M is the number of basic functions, x is the output data vector, W_k*j*_ is the connection weight between the basis function and the output layer, $\emptyset_{j}$ is the nonlinear function of unit *j* in the RBF, and W_k0_ is the weighted connection in the output layer.

#### 6.2.2. Adaptive Neuro-Fuzzy Inference Systems (ANFIS) Model

The adaptive neuro-fuzzy inference systems (ANFIS) model, proposed by Jang et al. (1993), is a hybrid architecture composed of fuzzy inference systems (FIS) enhanced with ANN features [92]. This model compensates for the disadvantages of fuzzy inference systems such as trial–error methods in tuning membership functions parameters, high time-consumption in design, a continuous and complete rule base, and a lack of standard methods for transforming human knowledge into a rule base [93]. On the other hand, the main advantages of the ANFIS model are the lack of any requirement for a mathematical model, its simulation of human thinking, and its simple interpretation of results [93,94].

The ANFIS architecture has five layers according to Takagi–Sugeno rules (Figure 11) [94]. The first layer (adaptive) forms the premise parameters. In the second layer, the products of the involved membership function are computed. Then, in the third layer, the sum of inputs is standardized. In the fourth layer, the adaptive if-node computes the contribution of if–then rules to the ANFIS output and forms the consequence parameters, and the fifth and final layer, sum all of the inputs [94,95].

In addition, a FIS structure consists of three main components, including a rule base, a database, and a reasoning mechanism [96]. The rule base has enough if–then rules for the scope of input variables, while the database determines membership functions applied in fuzzy rules, and the reasoning mechanism is responsible for an inference procedure [96]. Besides, the ANN portion in this architecture can improve membership functions related to the FIS structure based on its training mode, according to a training and checking dataset [93]. The training process is based on a hybrid learning algorithm or a BP algorithm. The hybrid learning algorithm defines premise parameters with a gradient method and consequence parameters with a least square model. With the BP algorithm, error signals are transmitted back, and new premise parameters are calculated using the gradient method.

In fact, Domanska and Wojtylak applied the ANFIS model to forecast concentrations of CO, SO_2_, and especially PM_10_ and PM_2.5_ pollution. Meteorological data (e.g., time horizon, weather forecast, meteorological situation, and pollution concentration) in Poland were defined as inputs. It was estimated that the ANFIS model forecasted PM_10_ and PM_2.5_ concentrations for 24–36 h with the lowest error value [97]. In another study, Polat and Durduran predicted daily air pollution levels including PM_10_ concentrations in Turkey using a combined ANFIS/output-dependent data scaling (ODDS) network model [98]. PM_10_ forecast was contributed by meteorological variables (e.g., temperature, humidity, pressure, and wind velocity attributes) as input data, as training data, and actual PM_10_ concentrations from the air quality statistics database of the Turkish Statistical Institute. The combined ANFIS/ODDS model was demonstrated to be an effective PM_10_ prediction method (Figure 12).

## 7. Limitation of Study

The limitation of this review is the relatively small number of individual studies evaluating personal exposure while passengers are waiting at bus stations, especially in heavily polluted countries. This fact could have been the cause of some negligible estimates.

## 8. Conclusions

This review identifies levels and factors affecting PM pollution at bus stations and calculates passenger exposure while waiting at bus stations as well as the health effects of such exposure. PM pollution at bus stations is primarily attributable to the combustion of fuel in vehicles, the wear of vehicle components (e.g., tires and brakes), the suspension of road dust, cigarette smoking, and industrial emissions. In addition, meteorological factors including wind speed, vehicle flow, temperature, relative humidity, pressure, rain volume, and cloudiness as well as the design and the direction of bus stations also influence distributions of PM inside or outside bus stops. It was proven that bus stops can accumulate high PM levels, thereby enhancing personal exposure to PM and leading to related diseases such as CVD, respiratory health effects, and diabetes. In order to accurately predict PM pollution, some predictive methods have been developed, typically ANN and ANFIS networks. In which, ANN is a powerful data modeling method with proven efficacy in solving complex problems in the fields of alignment, prediction, and classification, while the ANFIS model has several advantages such as non-requirement of a mathematical model, simulation of human thinking, and simple interpretation of results relative to other predictive methods. These networks, based on real PM concentration databases from local authorities, relative meteorological parameters, and some statistical software, endow PM forecasts with small error values. Consequently, development of air quality prediction networks at bus stops, which potentially can reduce health risks to a minimum, is both feasible and necessary. Future research should focus on evaluation of personal exposure during travels by public transportation, especially at bus stations.

## Figures and Tables

**Figure 1 ijerph-15-02886-f001:**
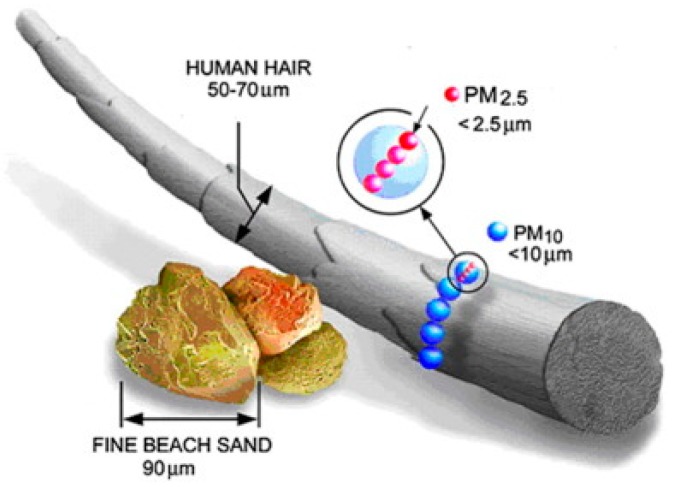
Size comparisons of particulate matter (PM) [12]. “Reproduced with permission from (Kim et al., Environment International) published by (Elsevier, 2015)”.

**Figure 2 ijerph-15-02886-f002:**
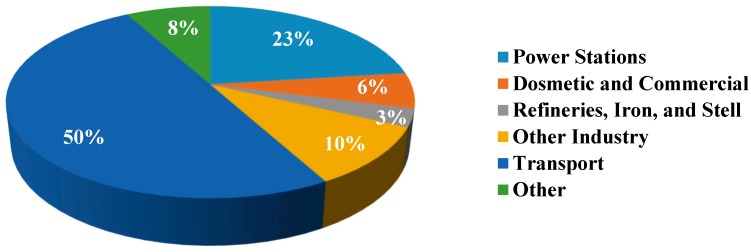
Sources of PM_10_ (coarse PM) pollution in UK (2001) [22].

**Figure 3 ijerph-15-02886-f003:**
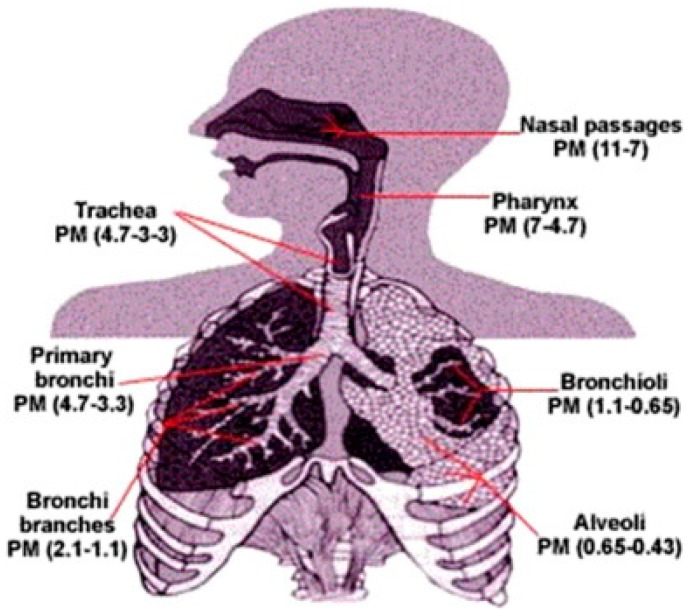
Potential for deposition of particles of different sizes [19]. “Reproduced with permission from Kim et al. (Environment International; published by Elsevier, 2015)”

**Figure 4 ijerph-15-02886-f004:**
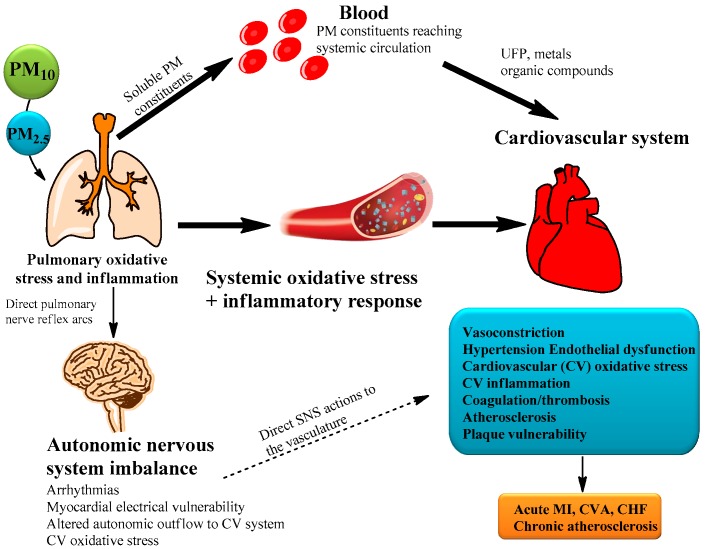
Biological pathways which PM can cause cardiovascular disease (CVD) [43]. (UFP: Ultrafine particle, MI: Myocardial infraction, CVA: Cerebrovascular accident, and CHF: Congestive heart failure).

**Figure 5 ijerph-15-02886-f005:**
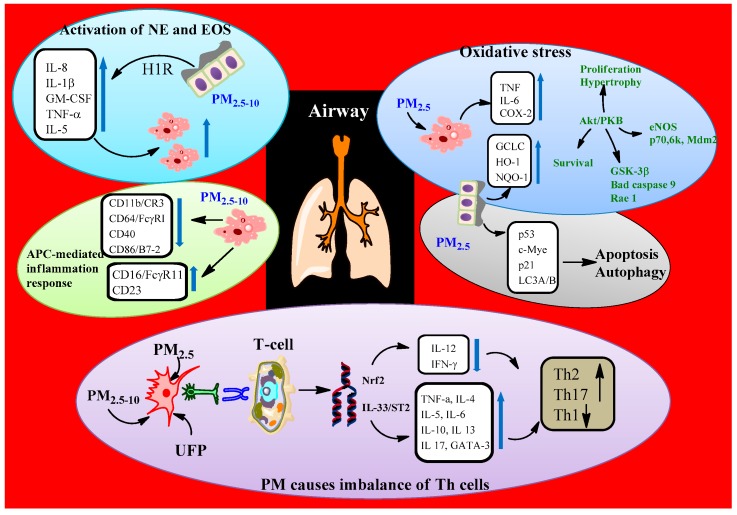
Mechanisms of PM’s effect in allergic respiratory diseases [51].

**Figure 6 ijerph-15-02886-f006:**
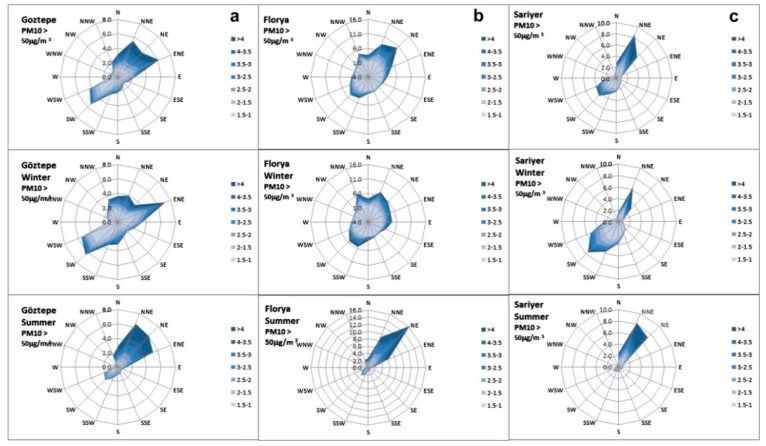
Wind speed increases annually in winter and summer seasons for (**a**) Goztepe, (**b**) Florya, and (**c**) Sariyer in Istanbul and their effects on classified PM_10_ levels [68]. “Reproduced with permission from Unal et al., (Atmospheric Environment; published by Elsevier, 2011)”.

**Figure 7 ijerph-15-02886-f007:**
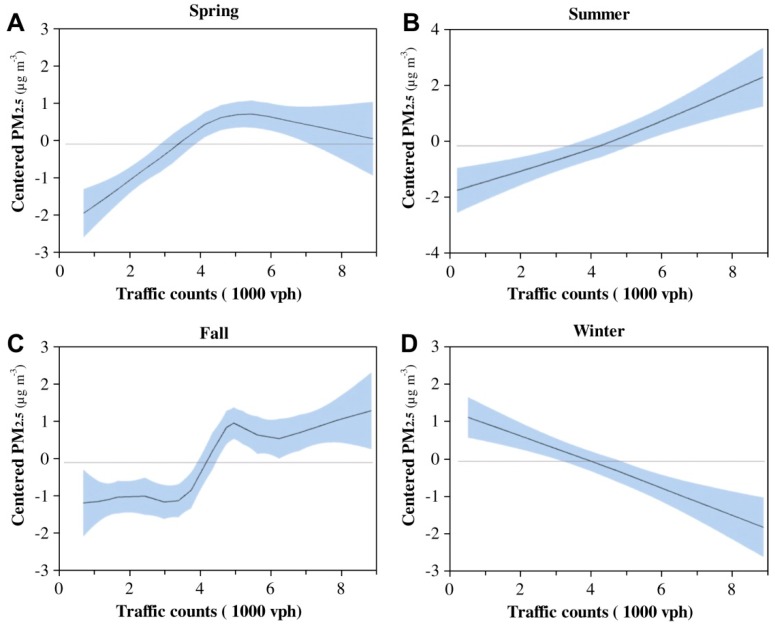
Relationship between PM_2.5_ concentrations and general traffic counts at major highway in Detroit, Michigan during summer season [10,72]. “Reproduced with permission from (Zhang and Batterman, Atmospheric Environment; published by Elsevier, 2010)”

**Figure 8 ijerph-15-02886-f008:**
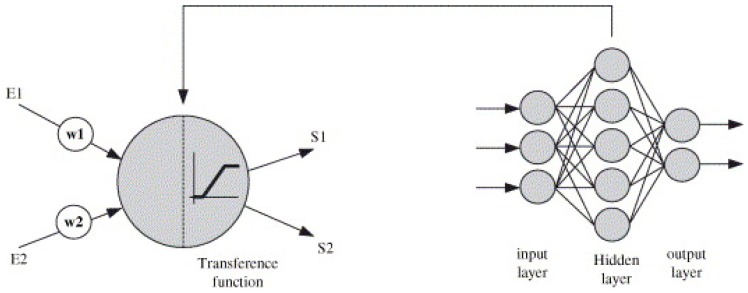
Artificial neural network (ANN) structure [82]. “Reproduced with permission from Ordieres et al., Environmental Modelling & Software; published by Elsevier, 2005)”.

**Figure 9 ijerph-15-02886-f009:**
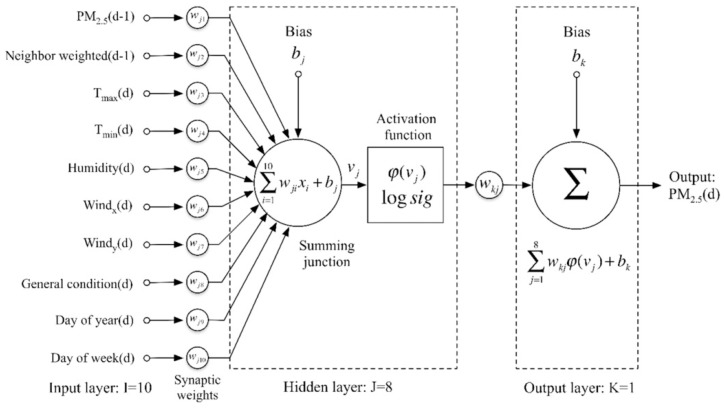
Architecture of Multilayer Perceptron (MLP)-type neural network [88].

**Figure 10 ijerph-15-02886-f010:**
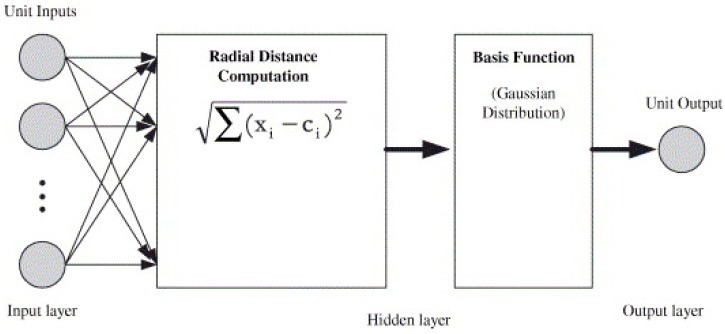
Radial basis function (RBF) structure [82]. “Reproduced with permission from Ordieres et al., Environmental Modelling & Software; published by Elsevier, 2005)”.

**Figure 11 ijerph-15-02886-f011:**
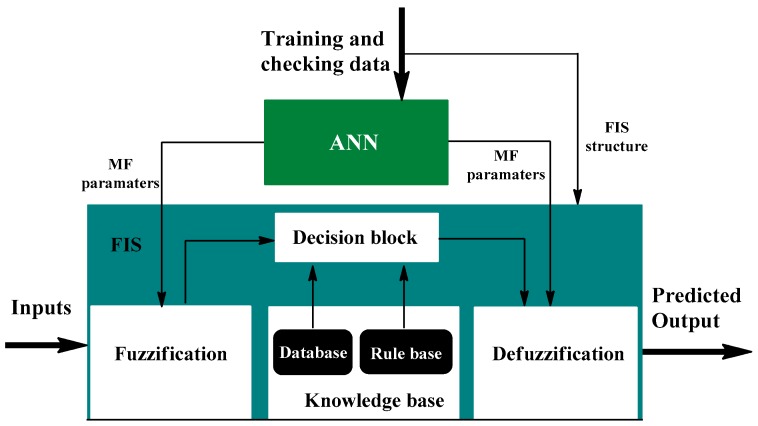
Adaptive neuro-fuzzy inference systems (ANFIS) structure [94].

**Figure 12 ijerph-15-02886-f012:**
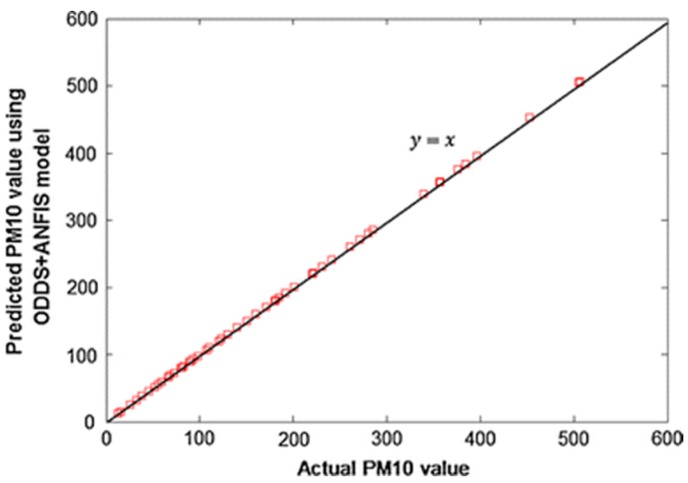
Predicted and actual PM_10_ concentration values and their relationship according to combined ANFIS/ output-dependent data scaling (ODDS) model [98]. “Reproduced with permission from Polat and Durduran, Neural Computing & Applications; published by Springer Nature, 2011).”

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
