# Peer review of "Particulate Matter Exposure of Passengers at Bus Stations: A Review"

_ijerph, 2018, doi:10.3390/ijerph15122886_

Round 1
Reviewer 1 Report
This study is comprehensive, relevant, and important. However, it is very poorly organized and has significant grammar, wording, and spelling issues. For example Line 349, the word "mush" is used instead of "much". While "mush" is an English word, which allowed the paper to pass a simple spell check, it's not the correct word. Therefore it is necessary for the paper to be reviewed and rewritten using the assistance of a native English speaker or someone who is fluent.
While the main focus of the paper is supposed to be exposure at bus stops, and the majority of it focuses on this theme, the inclusion of ANN and machine learning is very artificial. It seems that this section could be a separate manuscript or omitted, but as it stands it does not fit in well with the rest of the paper. A possible way to include it could be at the end of the paper in a section labeled "Future Directions" or something similar, but the current inclusion is strange and does not fit well at all. Furthermore, the organization can be improved to provide a more logical and smooth flow.
Author Response
First of all, thank you for your critical comments. The whole manuscript has been corrected, especially on English Grammar by a native speaker (vide infra).
In addition, PM pollution has been precisely forecasted using a number of artificial intelligence models such as artificial neural network (ANN) and adaptive neuro-fuzzy inference systems (ANFIS). Through the predictable values, passengers can avoid exposure to high pollution during their trips by checking current pollution values. Although the PM prediction method may not be well fit in this manuscript, it is an important section of our study so it cannot be excluded. Nevertheless, the manuscript has been re-organized again to improve the logical of content and smooth flow. Simultaneously, it is added a new section entitled “Future direction for reduction in personal PM exposure”, and the prediction models are presented in this section.
“6. Future directions for reduction in personal PM exposure
6.1. Pollution prevention and control
PM emissions can be minimized by pollution prevention and control technologies. Some typical regulations and air quality management actions have been applied to reduce pollution from concerning sources (traffic, industry, and human activities). Pollution prevention includes choosing clean fuels (natural gas) instead of diesel fuel, whose substitution can reduce the formation of secondary particles (NO2 and NH3) in traffic emission; utilizing cleaner processes such as advanced coal combustion technologies (coal gasification and fluidized-bed combustion) that may lower concentrations of products by incomplete combustion, and replacing older devices with cleaner ones enables better burning and fewer PM emissions [78,79]. Particularly, in order to reduce personal exposure indoors and outdoors, especially at bus stations, passengers should avoid smoking in bus shelters, reduce traveling during rush hour, and avoid outdoor activities when pollution levels are high [80].
In addition, a variety of PM-removal technologies with different physical and economic characteristics have been applied, such as impingement separators and electrostatic precipitators. Impingement separators rely on the inertial properties of the particles to separate them from the carrier gas stream, thereby collecting medium-size and coarse particles, while electrostatic precipitators show a high efficiency in collecting PM2.5 with well-designed, well-operated, and well-maintained systems, that can remove PM by using an electrostatic field to attract particles onto their electrodes [79].
6.2. Forecasting of PM pollution
In order to avoid high pollution exposure, commuters should check current and forecasted air quality levels before undertaking outdoor and public transport. Forecasting of pollution is an important tool for taking effective pollution control measures that can provide an early warning against harmful air pollutants [81]. Among these various predictive models, modes of artificial intelligence such as artificial neural network (ANN) and adaptive neuro fuzzy inference systems (ANFIS) models are classic methods used to forecast air pollution quantitatively [81,82].”

Reviewer 2 Report
Dear Authors,
This review paper is easy to read and is compact.
Sometimes, English language is felt strange, but maybe acceptable.
If possible, it may be better to review English language carefully.
Sincerely,
Author Response
Thank you for your kind comments. It has been tried to present a comprehensive overview of PM pollution at bus station. The whole manuscript has been corrected, especially on English Grammar by a native speaker (vide infra).

Reviewer 3 Report
The manuscript 'Particulate matter exposure of passengers at bus station: a review' is about a important subject of research. Nowadays, air pollution is one of the biggest hazards the world is facing. Air pollution can severely harm the health of people and some of the major health issues caused by air pollution are, e.g., cardiovascular diseases, respiratory diseases, even death in some cases.
In my opinion, this is a important topic of reseach and the manuscript is well writen and the methology is clear. The authors can be improve this manuscript in section 7 (Conclusions). I think that the authors can present the limitations of work done as well as the new topics to discuss for future researchs.
L49 - I suggest to authors insert a paragraph about the manuscript structure.
Author Response
Thank you for your positive comments. According to your suggestion, it is added a new section entitled “Limitation of study” as well as some subsequent topics for future researches in the Conclusion.
“7. Limitation of Study
The limitation of this review is the relatively small number of individual studies evaluating personal exposure while passengers are waiting at bus stations, especially in heavily polluted countries. This fact could have been the cause of some negligible estimates.”
“8. Conclusion
This review identifies levels and factors affecting PM pollution at bus stations and calculates passenger exposure while waiting at bus stations as well as the health effects of such exposure. PM pollution at bus stations is primarily attributable to the combustion of fuel in vehicles, the wear of vehicle components (e.g., tires and brakes), the suspension of road dust, cigarette smoking, and industrial emissions. In addition, meteorological factors including wind speed, vehicle flow, temperature, relative humidity, pressure, rain volume, and cloudiness as well as the design and the direction of bus stations also influence distributions of PM inside or outside bus stops. It was proven that bus stops can accumulate high PM levels, thereby enhancing personal exposure to PM and leading to related diseases such as CVD, respiratory health effects, and diabetes. In order to accurately predict PM pollution, some predictive methods have been developed, typically ANN and ANFIS networks. In which, ANN is a powerful data modeling method with proven efficacy in solving complex problems in the fields of alignment, prediction, and classification, while the ANFIS model has several advantages such as non-requirement of a mathematical model, simulation of human thinking, and simple interpretation of results relative to other predictive methods. These networks, based on real PM concentration databases from local authorities, relative meteorological parameters, and some statistical software, endow PM forecasts with small error values. Consequently, development of air quality prediction networks at bus stops, which potentially can reduce health risks to a minimum, is both feasible and necessary. Future research should focus on evaluation of personal exposure during travels by public transportation, especially at bus stations.”
In addition, at line 49, the paragraph has been upgraded with the detail structure of the manuscript.
“This present review provides an overview of PM pollution at bus station, focusing on the characteristics, health effects, and factors affecting PM as outlined in the following sections below: “Classification and sources of PM”, “Health problems caused by PM exposure”, and “Factors affecting PM pollution at bus stations”, respectively. In addition, the personal exposure levels to pollutants around the world (Europe, the Americas, and Asia) are explicated in the section “Personal exposure to PM at bus stations”. Next, in the section “Future directions for reduction in personal PM exposure”, some PM-pollution predictive methods are proposed. Finally, a limitation and conclusions are given in the “Limitation of study” and “Conclusion” sections, respectively.”

Round 2
Reviewer 1 Report
The manuscript is much improved.